# Provision of Dental Care to Indigenous South Australians and Impacts on Improved General Health: Study Protocol

**DOI:** 10.3390/ijerph20042955

**Published:** 2023-02-08

**Authors:** Lisa Jamieson, Joanne Hedges, Zell Dodd, Priscilla Larkins, Cindy Zbierski, Sonia Nath, Kostas Kapellas, Xiangqun Ju

**Affiliations:** 1Adelaide Dental School, The University of Adelaide, Adelaide 5000, Australia; 2Umoona Tjutagku Health Service, Coober Pedy 5723, Australia; 3South Australia Health, Adelaide 5000, Australia; 4Ceduna Koonibba Aboriginal Health Service, Ceduna 5690, Australia

**Keywords:** Indigenous South Australian, culturally safe dental care, chronic diseases

## Abstract

Background: Indigenous South Australians carry a disproportionate burden of dental diseases, with approximately 80 percent of Indigenous adults having both periodontal disease and dental caries. The chronic inflammatory nature of many dental conditions means there are widespread systemic impacts, particularly on type 2 diabetes, chronic kidney disease and cardiovascular disease. Evidence suggests there are barriers experienced by Indigenous South Australians in accessing timely and culturally safe dental care. This study aims to: (1) elicit the views of Indigenous South Australians regarding their perspectives of what comprises culturally safe dental care; (2) provide such dental care and; (3) assess any changes in both oral and general health using point-of-care testing following receipt of timely, comprehensive and culturally safe dental care. Methods/Design: This mixed-methods study will involve qualitative interviews and an intervention without randomisation. The qualitative component will comprise seeking perspectives of Indigenous South Australians regarding what culturally safe dental care means for them. For the intervention component, participants will take part in oral epidemiological examinations at baseline and 12-month follow-up (after receipt of dental care), which will include collection of saliva, plaque and calculus, as well as completion of a self-report questionnaire. The primary outcome measures—changes in type 2 diabetes (HbA1c), cardiovascular disease (CRP) and chronic kidney disease (ACR)—will be obtained by blood/urine spot from a finger prick/urine collection at baseline and 12-month follow-up via point-of-care testing. Results: Participant recruitment will commence in July 2022. The first results are expected to be submitted for publication one year after recruitment begins. Discussion: The project will have a number of important outcomes, including increased understanding of what culturally safe dental care means for Indigenous South Australians, the delivery of such care, and empirical evidence of how culturally safe dental care leads to better prognosis for chronic diseases linked with poor oral health. This will be important for health services planning, especially in the Aboriginal Community Controlled Health Organisation sector, where the management of dental diseases in a culturally safe manner for better chronic disease outcomes is currently insufficiently understood, planned and budgeted for.

## 1. Background

Oral health is an important indicator of overall health and wellbeing [1]. The two most common dental conditions are periodontal (gum) disease and dental caries (tooth decay). Periodontal disease is an inflammatory and infectious condition of the supporting bone and soft tissues around teeth. Clinical characteristics include gingival bleeding, receding gum tissues and tooth mobility [2]. Dental caries arises from prolonged demineralisation of tooth structures resulting from carbohydrate metabolism by acidogenic bacteria [3]. When left untreated, dental caries can cause substantial pain and serious infections that may spread to other areas of the head and neck [1]. The prevalence of untreated dental caries and periodontal disease is higher among Indigenous Australians relative to non-Indigenous Australians, with a higher proportion of Indigenous Australians not receiving preventive dental care [4]. Indigenous Australians tend towards unfavourable dental visiting patterns, broadly associated with accessibility, cost and a lack of cultural awareness by some service providers [5]. Direct impacts of poor oral health include pain, functional impairment and aesthetic concerns. Indirect impacts include difficulties eating, chronic inflammation, systemic infection, links to comorbidity and delays in kidney transplant waitlisting [4].

### 1.1. What Is the Link between Oral Health and General Health?

The mouth contains over 300 species of bacteria, with an entry point to the digestive, respiratory and circulatory systems [6]. Evidence shows that oral bacteria and the inflammation associated with oral bacteria-related diseases is indisputably linked with systemic health and disease [7,8,9]. Such as increasing C-reactive protein (CRP) levels (Figure 1). Certain diseases, such as type 2 diabetes, can lower the body’s resistance to infection, making oral health problems more severe. Periodontal disease has been described as the 6th complication of type 2 diabetes [10], with the relationship understood to be bi-directional [11].

### 1.2. What Is the Mechanism by Which Poor Oral Health Affects General Health?

There are two main mechanisms by which poor oral health, specifically periodontal disease, contributes to systemic disease. The first is a direct mechanism; as chronic periodontitis progresses, the epithelium lining periodontal pockets becomes ulcerated, providing a direct entry point for periodontal bacteria into the systemic circulation. The circulating bacteria may directly affect certain organs, for example, periodontal bacteria have been detected in thrombi from patients with acute myocardial infarction suggesting a role in the pathological changes that occur in atheromatous plaques [12]. Periodontitis additionally represents an indirect source of inflammation that is a significant contributing factor in the pathogenesis of systemic conditions (currently 57 identified) [13], including cardiovascular disease, type 2 diabetes, chronic kidney disease, rheumatoid arthritis, respiratory disease, obesity, metabolic syndrome, cancer, adverse pregnancy outcomes and neurogenerative disorders [14]. The level of CRP in the blood is an accepted method of measuring systemic inflammation in individuals, with robust evidence that CRP levels are elevated in clients with periodontitis [15].

### 1.3. What Are the Mechanisms by Which Improved Oral Health Impacts General Health?

In a field of research that has over 30 years of clinical trials, there is conclusive evidence that comprehensive dental care improves systemic diseases, specifically through scaling and root planing of the teeth and gums where periodontal disease is evident. This serves to reduce the bacterial load in the mouth, reduce the inflammatory burden contributed by the oral cavity to the systemic system, and provide a smooth tooth surface that is difficult for bacteria to adhere to in future [16]. In a recent systematic review and meta-analysis on the impact of the treatment of periodontitis on systemic health, Orlandi and colleagues [17], using evidence from 47 clinical trials, reported that treatment of periodontal disease after 6–12 months lowered systemic inflammation (reduction in hs-CRP and IL-6) and improved metabolic control (reduction in glucose level) and endothelial function (increase in brachial artery flow-mediated dilatation). Other evidence demonstrates how management of periodontitis in patients with systemic conditions not only creates substantial public health benefits but is cost-effective, reducing the costs associated with systemic disease complications [18,19]. Orlandi et al. concluded that, ‘given the high prevalence of periodontal disease at a population level, implementing oral health strategies as a vehicle of decreasing systemic inflammation burden should not be underestimated and considered in public health strategies worldwide’ [17].

### 1.4. Why Is the Provision of Culturally Safe Dental Care Important?

Many Indigenous Australians do not seek dental care because of a lack of cultural safety in the dental setting [20,21]. There is, for many, a deep mistrust of dental service providers who, up until the 1990s, were authorised to remove all remaining teeth of Aboriginal Australians as a condition for surgery for heart disease [4]. The provision of dental care for Indigenous Australians essentially adopts a Western model which, especially in the private sector, is profit-driven and very much favours multiple appointments per day, zero tolerance for cancellations/no shows and discouragement of extra family/friends to attend in a support capacity. There is generally scant regard for culturally safe care, with many dental clinics using uber high technology equipment and facilities, but not having simple things, such as Reconciliation Action Plans or Indigenous art/reconciliation statements on clinic walls/reception areas [22]. There is often limited experience of all dental staff in working with clients with substantial dental fear/anxiety [23]. All these factors act as deterrents for many Indigenous Australians to seek timely, and regular, dental care.

### 1.5. Why Would the Provision of Culturally Safe Dental Care Have Impacts on General Health?

If dental services provided to Indigenous Australians are not culturally safe, Indigenous Australians will not present for dental care. If Indigenous Australians do not present for dental care, the impacts of poor oral health may manifest substantially on the 57 systemic conditions identified as being associated with periodontal disease. Being able to provide evidence of the benefits of improved oral health among Indigenous Australians for improved systemic health is essential before there can be any translation to health policy. This includes, in the context of Indigenous Australians, having dental included in chronic disease health management plans, and regular engagement of health staff managing Indigenous patient chronic diseases with dental staff.

### 1.6. What about Confounding?

One of the main difficulties in studying links between periodontitis and systemic disease is that they share some risk factors. Any changes in systemic disease biomarkers will be influenced by a multitude of factors, not just treatment of dental disease. These include medical management, comorbidities, new medications, adherence to medication, health literacy and education. Traditionally, randomised controlled trials (RCTs) have been the only study design to identify cause and effect relationships that account for such confounders. For research questions (such as those from this project) in which conducting RCTs are unethical or impractical, causal inference methods can be used to estimate causal effects from observational data [24]. To estimate the causal effects of dental care on selected biomarkers (HbA1c, CRP and ACR) among Indigenous populations, this research project will adjust for multiple confounders including sociodemographic characteristics (e.g., education, income, health literacy), comorbidities, history of and current medication use, among other factors. We will implement cutting-edge sensitivity analysis methods to investigate the impact of the unmeasured confounders on the causal effect estimate [25]. Sensitivity analysis methods indicate how strong the unmeasured confounders would need to be to explain away the observed causal effect, elucidating whether unmeasured confounding had an impact (if any) on the causal effects estimated [26]. The use of sensitivity analysis to investigate the impact of unmeasured confounding on causal effects estimated from observational data is best practice in epidemiological studies [27] and will be incorporated in this research project.

### 1.7. Feasibility

Our team has 18+ years working in partnership with Indigenous communities in Australia providing dental interventions, including the collection and assessment of systemic disease biomarkers. For example, Jamieson was the dentist involved in the Aboriginal Birth Cohort study in the Northern Territory, in which blood and urine was collected from children from the age of 4 years [28]. Kapellas and Jamieson were involved in two interventions in the Northern Territory involving comprehensive dental care for Indigenous Australians. Both studies included collection of blood and urine to assess clinical biomarkers before and after the dental intervention [29,30,31]. In a large study involving 1011 Indigenous South Australians in an oral HPV and oropharyngeal cancer study, and involving Hedges and Jamieson, retention of 75% of participants was obtained at 12-month follow-up (*n* = 749), and again at 24-month follow-up (*n* = 741) [32]. Our team also has substantial experience in the implementation, collection and analysis of qualitative data around dental service provision from the lens of Indigenous Australians [20,33]. This evidence demonstrates that our proposed objectives are feasible and that we are the best team to address the study aims.

### 1.8. What Is Point-of-Care Testing?

Point-of-care testing is a quality-assured pathology service using analytical devices provided near to the patient rather than in the traditional environment of a clinical laboratory [34]. The objective of point-of-care testing is to minimise the time to obtain a test result, and to avoid taking a large volume of testing material (typically blood or urine). There are a number of point-of-care testing analytical devices available on the market for a large range of systemic disease biomarkers. Sensitivity analyses indicates that point-of-care results are comparable with those obtained from third party laboratory processing [35]. Point-of-care testing is especially advantageous when working in partnership with Indigenous Australian communities, particularly those living in rural and remote locations, as results can be immediate, there is no involvement of third-party laboratory services, smaller volumes of biodata is required (compared to traditional laboratory testing) and specimens do not need to be stored/maintained over the long-term.

### 1.9. Study Aims

Our study aims are as follows:

Aim 1: To determine barriers to, and facilitators of, culturally safe dental care among Indigenous South Australians. Hypothesis: Barriers to culturally safe dental care will include a range of macro (dental system organisation) and micro (individual dental providers/services) factors.

Aim 2: To work with the Indigenous South Australian community and dental service providers to provide culturally safe dental care.

**Hypothesis** **1**.*Culturally safe dental care can be provided with sufficient training, resources, goodwill and commitment from the broader dental community*.

Aim 3: To ascertain changes in both clinical and self-reported oral health and general health following completion of dental care, using point-of-care testing for the general health biomarkers.

**Hypothesis** **2**.*Receipt of culturally safe dental care will contribute to improvements in dental health, clinical biomarkers of systemic disease as ascertained through point-of-care testing, and self-reported general and oral health*.

## 2. Method/Design

This mixed-methods study will follow recommendations of Creswell [36] and will comprise 2 components (Figure 2). The qualitative component will seek perspectives of Indigenous South Australians on how they define culturally safe dental care. In the intervention component, participants will take part in oral epidemiological examinations at baseline and 12-month follow-up (after receipt of dental care), which will include collection of saliva, plaque and calculus, and completion of a self-report questionnaire. For adults, blood and urine will also be collected at baseline and 12-month follow-up to assess biomarkers of systemic conditions using point-of-care testing.

### 2.1. Study Population, Recruitment and Retention

We will recruit 1000 Indigenous South Australian children and 1000 Indigenous South Australian adults, with a focus on our key partner sites of Adelaide, Ceduna, Coober Pedy, Whyalla and Port Lincoln. Census data indicates approximately 22,000 Aboriginal adults and children reside in these areas. The investigators have an 18-year relationship with key Indigenous stakeholder groups in these locations. Recruitment strategies will be based on those successfully implemented in our past Indigenous dental projects. Participants will primarily be sourced from the participating Aboriginal Community Controlled Health Organisations (ACCHOs) in the first instance. Information about the study will be provided to all ACCHO staff (including receptionists, drivers, Aboriginal health workers/practitioners, doctors, nurses, auxiliary staff), through a presentation and one-on-one conversations. ACCHO staff who work closely with Indigenous clients (specifically Aboriginal health workers/practitioners, doctors and nurses) will, in turn, describe the study to potential participants and direct the study’s Indigenous research officers to facilitate recruitment if participants are willing. We will employ a snowball recruitment technique also; whereby study participants can contact their own family and friends to become involved in the study if interested. Retention strategies will involve: (1) employing staff who are committed to following up participants despite challenges in doing so; (2) ensuring participants are contacted on a regular basis to check accuracy of contact details; (3) maintaining relationships with appropriate stake holders; (4) ascertaining contact details of three key personnel who may know the whereabouts of participants should the study team be unable to contact them; (5) sending birthday and Christmas cards to participants; and (6) facilitating one-on-one relationships between study staff and participants, with study staff ideally seeing each of their participants for each phase of the research.

### 2.2. Inclusion and Exclusion Criteria

Participants will be aged 6 months and above, identify as being Aboriginal or Torres Strait Islander and planning to live in South Australia for the next 3 years. Participants, who are not enrolled during the original recruitment period, will not be eligible to participate in the 12-month follow-up.

### 2.3. Ethical Approval

Ethical approval has been obtained by the Aboriginal Health Council of South Australia’s Human Research Ethics Committee (04-22-990) and the University of Adelaide Human Research Ethics Committee.

### 2.4. Measures

*Primary outcome:* For purposes of the intervention component of the study, the primary outcome will be clinical biomarker changes of three systemic conditions assessed through point-of-care testing; type 2 diabetes (HbA_1c_), cardiovascular disease (C-reactive protein; CRP) and chronic kidney disease (albumin:creatinine ratio; ACR).

*Secondary outcomes*: Secondary outcomes will include dental caries experience, periodontal status, oral health-related quality of life, general health-related quality of life and other systemic disease biomarkers; IL-6, GFR, HDL and LDL cholesterol and triglycerides.

*Covariates:* Covariates will comprise socio-demographic factors (including education), oral health and general health behaviours (tobacco smoking, alcohol consumption), health literacy, medical management, medications, adherence to medication and comorbidities, especially obesity.

### 2.5. Data Collection Techniques

Qualitative, self-report and clinical information will be gathered in this study.

*Qualitative:* At baseline, participants, or carers of child participants, will be asked to take part in one-on-one interviews with study stuff to ascertain views on what culturally safe dental care means. At 12-month follow-up, after receipt of dental care, participants will take part in a follow-up interview to capture participant views on the dental care journey. Interviews will follow a standard prompt guide, will be audio-recorded and professionally transcribed.

*Self-report:* Self-report information pertaining to socio-demographic, oral health and general health factors will be gathered at baseline and 12-month follow-up. Socio-demographic factors will include age, sex, education level, employment, number of people who stayed in house the previous night and car ownership. Oral health factors will include self-rated oral health, oral health-related behaviours and quality of life. General health factors will include behaviours such as smoking and alcohol consumption and general health-related quality of life.

*Clinical:* At baseline, each participant will undergo an oral epidemiological examination and provide saliva, calculus and plaque samples. Adults will also provide blood drops (obtained via finger prick) and urine for point-of-care testing for HbA1c, CRP and ACR. This will be repeated at 12-month follow-up, after receipt of the culturally safe dental intervention.

Oral epidemiological examination: This will include tooth presence, dental caries experience, periodontal destruction, gingivitis and calculus, plaque, oral mucosal lesions and dental trauma.

Plaque, calculus and saliva samples: Saliva samples will be collected using commercially available kits, which will involve participants: (a) not eating or drinking for 30 min prior to collection; (b) removing the funnel lid on the container; (c) spitting until 2 mL reaches the fill line (takes 2 min); (d) closing the lid and unscrewing the lid from the funnel and; (e) placing the small cap on the tube and shaking the tube for 5 s. Dental personnel collecting oral epidemiological information will collect supragingival plaque and calculus samples using a sterile Gracey curette. The plaque and calculus will be collected from the buccal surface of two incisors and the mesio-buccal surface of the maxillary first and second molars, and placed into individual Eppendorf tubes. Saliva, calculus and plaque samples will be transported to a University of Adelaide laboratory for analysis.

Systemic disease biomarkers (adults only): A blood sample will be collected via a capillary finger stick with disposable lancet. Urine will be collected by a standard urine pot with sealed lid (and concealed in a paper bag). This will occur at baseline and 12-month follow-up. Three to five blood/urine spots will be analysed according to the directions provided for the point-of-care testing system Afinion™ for the analysis of primary outcomes HbA1c, CRP and ACR, and secondary outcomes IL-6, GFR, HDL and LDL cholesterol and trigylcerides.

### 2.6. Sample Size and Loss-To-Follow-Up

Given our past experience of large-scale dental initiatives involving Indigenous persons in South Australia [33], we anticipate retaining 75% of participants (750 adults and 750 children) at the 12-month follow-up. Based on evidence from the literature from other Indigenous Australian populations [33,34], and accounting for the sensitivity analysis requirements [30], this will give sufficient power to estimate the impacts of a culturally safe dental intervention on systemic disease biomarkers assessed through point-of-care testing, as well as other sub-group analyses.

### 2.7. Provision of Culturally Safe Dental Care

In 2020, the Australian Health Practitioner Regulation Agency, with whom all dental personnel in Australia need to be registered, stated that ‘*To ensure culturally safe and respectful practice, health practitioners must acknowledge colonisation and systemic racism, social, cultural, behavioural and economic factors which impact individual and community health’*. Key points, which will be utilised in the current study, include genuine engagement of Indigenous Australians in oral health care programs in environments that are culturally acceptable and sensitive. The investigators, through leadership of the study’s Senior Indigenous Research Officer (Hedges), will ensure all study activities are conducted in culturally safe ways. All non-Indigenous staff/students will undertake cultural safety training. Culturally safe practise will be reinforced through monthly debriefing sessions, organised by the study’s Senior Indigenous Research Officer to ensure that the many fears of Indigenous clients in accessing dental care (fear of being judged, fear of not being spoken to with respect, fear of child removal) are acknowledged and mitigated in personal and tailored ways. Through all aspects of data collection and dental service provision, Indigenous participants need to feel safe in any encounter with the dental sector; from the receptionist through to the specialist.

The clinical aspects of the oral health intervention will be based on the technique described by Tonetti and colleagues [37], and occur across a series of visits for each participant, based in each ACCHO’s unoccupied dental clinic or public dental clinics by registered oral health professionals who will receive additional training from the dentally qualified study staff, who have extensive expertise in this area. This involves intensive removal of subgingival dental plaque biofilms by scaling, root-planing and removal of teeth that cannot be saved, following administration of local anaesthesia. It will additionally involve the removal of all active dental caries in the dental hard tissues, including replacement of insufficient restorations, and comprehensive prophylactic cleaning with fluoride varnish. Dental care will commence immediately after the baseline visit. At no time will there be any costs incurred for study participants.

### 2.8. Data Analysis

Interpreting the results of both quantitative and qualitative methodologies will be conducted in accordance with Creswell’s recommendations for mixed-methods research [37]. In brief, the analysis plan for each aim is:

*Aim 1: Barriers/facilitators culturally safe dental care.* A two-stage approach to coding will be used in the thematic analysis of transcripts from the interviews. A primary analytical framework (the first stage) will be structured using the interview guide to create a coding scheme [38], which will group and categorize according to questions asked from the interview script. Developing thematic codes will be the 2nd step, which will more deeply reveal the assumptions, perceptions and conceptual constructs that undergird participants’ perceptions of what constitutes culturally safe dental care.

*Aim 2: Provision of culturally safe dental care.* The assessment of provision of culturally safe dental care will be through the perspective of participants who received the intervention; both children and adults. This will be garnered through qualitative interviews at the follow-up assessments, with the same analytical frameworks utilised as described for the analysis of Aim 1.

*Aim 3: Changes in clinical biomarkers.* Analysis of covariance will be performed to determine differences in the primary, secondary and other outcomes between baseline and 12-month follow-up. Causal inference and sensitivity analyses will be used to emulate a target trial approach [39]. This involves specifying the ‘target trial’ that would have been implemented in ideal circumstances (i.e., the impact of a culturally safe dental intervention on systemic disease biomarkers), then analysing the data in a way that emulates this as closely as possible, accounting for confounding and data missing to follow-up. Target trial emulation is an important step for identifying if an RCT is likely to have found a significant effect.

## 3. Results

Participant recruitment for this study will commence July 2022. We anticipate the first results to be submitted for publication one year following initial recruitment.

## 4. Discussion

Culturally safe dental interventions for Indigenous Australians to improve systemic disease biomarkers are essential to inform policy translation that might result in dental care provision as a part of chronic disease management in the Aboriginal Community Controlled Health Organisation setting across Australia. Point-of-care testing is an important element of intervention research involving the collection of biodata because it alleviates involvement of third-party laboratory services, means no storage is required, provides participants with immediate results and requires much smaller volumes of blood and urine. Understanding Indigenous Australian perspectives of what culturally safe dental services are will provide valuable knowledge for the creation of culturally safe training programs both in dental teaching institutions and in continuing professional development modules for dental professionals. The project provides a key example of how carefully calibrated, data-driven disease models can integrate with Aboriginal community views and expectations to estimate disease burden, deliver culturally safe dental care and guide policy decisions for more equitable resource distribution that includes dental as part of chronic disease management for Indigenous Australians.

The strengths of the study include it being the first to obtain and link all the information around oral health and general health through a mixed-methods, culturally safe intervention approach in partnership with the Indigenous South Australian community. The focus on engagement, enabling Indigenous community and individual preferences to play a large role in where and how data collection will take place, and how findings will be disseminated builds upon a platform of reciprocal and respectful research processes across almost 20 years [40]. The findings will also provide evidence of the important role Aboriginal Community Controlled Health Organisations play in linking the oral health–general health praxis to increase understanding at the broader community level of how improved oral health contributes to overall improved general health. Our work will support the recommendations of Poirier and colleagues [41], who suggested that Indigenous leadership in oral health has potential to eliminate many of the chronic disease challenges Indigenous communities face, with an example being expansion of the role of Indigenous Health Workers to provide basic oral health screening, prevention and care. The study has strong Aboriginal representation and leadership (half the authors are Aboriginal), with the study recognized as being a contribution to the health and wellbeing of Australia’s first peoples.

The study limitations include the sample frame for dental disease assessment and provision of culturally safe dental care being pragmatic, that is, utilising convenience sampling methodology, which maybe. there is a risk of sampling bias. While the efforts required to obtain a representative sample are recognised as being both expensive and time-consuming [42]. We remain committed to ensuring culturally safe dental care is provided to all study participants, informed by the findings from the qualitative analysis, and to translating study outcomes indicating the impacts on general health biomarkers resulting from a dental intervention for immediate policy translation. The study team will also work closely with dental clinics and be supported by the SA Dental Aboriginal Health Promotion team if client support is required.

Engaging dental clinics at the beginning of the project ensures they are familiar with the study team, the study’s aims and develops close ties/relationships to support the client. Asking questions of the study team eliminates unnecessary concern by the client, e.g., financial queries.

## 5. Conclusions

The project will have a number of outcomes, which is important for health services planning, and for Aboriginal health worker and patient education.

## Figures and Tables

**Figure 1 ijerph-20-02955-f001:**
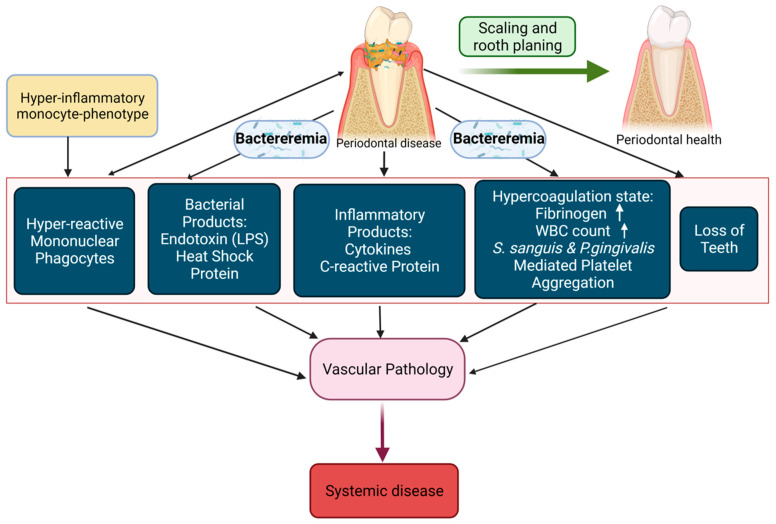
Scheme showing causal pathway of oral infections and systemic disease (‘🠙’ means ‘increased’).

**Figure 2 ijerph-20-02955-f002:**
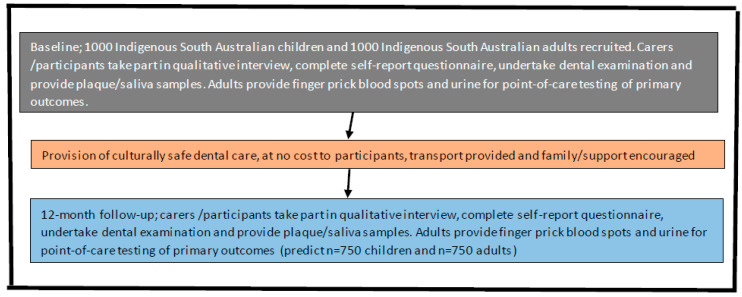
Study plan schema.

## Data Availability

The datasets generated and/or analyzed during the current study are not publicly available due to privacy issues of the participants. Data are available from the corresponding author on reasonable request.

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
