# Peer review of "Provision of Dental Care to Indigenous South Australians and Impacts on Improved General Health: Study Protocol"

_ijerph, 2023, doi:10.3390/ijerph20042955_

Round 1
Reviewer 1 Report
The manuscript offers an exciting topic to study ethnicity and its relationship with oral health. My main recommendation is regarding to the use of a mixed-methods approach. In this sense, I recommend that the authors define what kind of mixed methods design should be appropriated as proposed by John Creswell.
It could be important to define the strategies for combining/interpreting the results of both methodologies.
Author Response
Thank you for this recommendation. We have now cited Creswell’s suggestions of mixed-methods research (Line 191). We have also defined strategies for interpreting results of both quantitative and qualitative methodologies (Line 324).
Reviewer 2 Report
This manuscript addresses a protocol to: (1) elicit the views of Indigenous South Australians regarding their perspectives of what comprises culturally safe dental care; (2) provide such dental care and; (3) assess any changes in both oral and general health using point-of care testing following receipt of timely, comprehensive and culturally safe dental care.
It is nesessary for understand health services planning, especially in the Aboriginal Community.
・Method
I think a bias how to recruit paticipants :
We will employ a snowball recruitment technique also; whereby study participants can contact their own family and friends 215 to become involved in the study if interested.
Author Response
There will be a bias in recruiting participants, but it is a convenience study, not a representative one. Amendments made to the text to indicate limitations around bias in recruitment (Line 389).
Reviewer 3 Report
This protocol is interesting, well written and of high relevance.
There is one major problem, and this is the reason why i Suggested major revision even if the revision might be technically minor.
Who is the representative of the Aboriginal population in the study team? This representative (one ore more) should co-author the protocol, participate in steering the study and co-author the publiction of results (and so, first of all, participate in the interpretation of data).
The crucial issue is that it should be made clear that this research is considered as a contribution to improvement of the life of indigenous people, not an intrusion to theit traditional life.
Therefore, if indigenous representatives are visible in the steering board and the author team of this study, I will warmly recommend publication. If not, then I would come to the conclusion that this is once more research of white people on others without equal participation of the others. Therefore, please clarify this issue.
Author Response
Joanne Hedges, Zell Dodd, Priscilla Larkins and Cindy Zbierski are representing the Aboriginal population in the study team and will be undertaking all that the reviewer suggests. Addition to text that the study is a contribution to the health and wellbeing of Indigenous Australians (Line 386).
Round 2
Reviewer 3 Report
I am happy to read that four authors are Aboriginal, and hence, the Aboriginal population is not only an object of research; people are studying their own matter. This is the kind of research we need in that area. The presentation of the study protocol is fine. I recommend acceptance as is.